# Impact of Different Reanalysis Data and Parameterization Schemes on WRF Dynamic Downscaling in the Ili Region

**Yulin Zhou and Zhenxia Mu \***

College of Water Conservancy and Civil Engineering, Xinjiang Agricultural University, Urumqi 830052, China; zhouyulin19921103@126.com

**\*** Correspondence: muzhenxia@126.com; Tel.: +86-139-9922-7089

**Abstract:** Different reanalysis data and physical parameterization schemes for the Weather Research and Forecasting (WRF) model are considered in this paper to evaluate their performance in meteorological simulations in the Ili Region. A 72-hour experiment was performed with two domains at the resolution of 27 km with one-way nesting of 9 km. (1) Final Analysis (FNL) and Global Forecast System (GFS) reanalysis data (hereafter, WRF-FNL experiment and WRF-GFS experiment, respectively) were used in the WRF model. For the simulation of accumulated precipitation, both the WRF-FNL (mean bias of 0.79 mm) and WRF-GFS (mean bias of 0.31 mm) simulations can display the main features of the general temporal pattern and geographical distribution of the observed precipitation. For the simulation of the 2-m temperature, the simulation of the WRF-GFS experiment (mean warm bias of 1.81 °C and correlation coefficient of 0.83) was generally better than that of the WRF-FNL experiment (mean cold bias of 1.79 °C and correlation coefficient of 0.27). (2) Thirty-six physical combination schemes were proposed, each with a unique set of physical parameters. Member 33 (with the smallest mean-metric of 0.53) performed best for the precipitation simulation, and member 29 (with the smallest mean-metric of 0.64) performed best for the 2-m temperature simulation. However, member 29 and 33 cannot be distinguished from the other members according to their parameterizations. For this domain, ensemble members that contain the Mellor–Yamada–Janjic (MYJ) boundary layer (PBL) scheme and the Grell–Devenyi (GD) cumulus (CU) scheme are recommended for the precipitation simulation. The Geophysical Fluid Dynamics Laboratory (GFDL) radiation (RA) scheme and the MYJ PBL scheme are recommended for the 2-m temperature simulation.

**Keywords:** WRF; different reanalysis data; physics parameterization; Ili Region

## 1. Introduction

Precise flood and drought forecasts are particularly difficult due to the scarcity of observation data and the complex topography over the Ili Region [1]. To improve runoff predictions, accurate and high-resolution meteorological data are essential. To date, no dynamic downscaling model studies have been conducted to record high-resolution climate data in the region. Thus, a WRF model study of the hydrological climate of the Ili Region is needed.

As a mesoscale downscaling technique, the WRF model has been widely used in both studying and forecasting a variety of meteorological events [2]. Many studies [3–5] have indicated that many climatic factor forecasts (such as those of heavy rainfall, maximum and minimum temperature, and wind speed) can be improved using WRF ensemble model techniques. The WRF mesoscale model systems can provide useful probability information for rainfall and runoff forecasting [6–8]. Therefore, based on the existing literature, the regional model performs quite well in this area.

However, many questions remain, including the selection of various physical schemes and boundary conditions (LBCs) and decisions on the domain size and resolution, which challenge the WRF model's attempts for precise simulation. (1) For the sensitivity analysis of various physical schemes on the WRF model, the simulation of mesoscale convective system (MCS) characteristics is highly sensitive to the parameterization scheme choice over southeast India [9]. The model provided better forecasting of heavy rainfall events using the logical combination of Goddard microphysics, Yonsei University (YSU) PBL, and Noah land surface model (LSM) schemes over India [10]. The CU and PBL schemes have important implications for the simulation of the rainfall in Andalusia, while there is no significant difference between microphysics (MP) schemes [11]. There are different choices for the physics and dynamics of the WRF model, enabling users to optimize the model for specific geographies [12]. As the number of parameterization schemes increases, it becomes more and more difficult to determine the optimal combination of physical parameterization schemes. Therefore, the selection of various physical schemes is of great significance for the simulation of climatic factors. (2) For studies on the evaluation of different initial boundary conditions and LBCs in the WRF model, reanalysis data, such as NCEP [13,14], FNL [15], CMIP5 [16], GFS [17], ERA-40 [18], and NARR [19], are widely used to validate the performance of the WRF model because they minimize the error of LBCs. The WRF model is simulated using different reanalysis data, and the results show that the WRF model has significant differences in precipitation and temperature. [20]. Therefore, selecting different LBCs is necessary for estimates over the Ili Region. (3) For the impact of different horizontal and vertical resolutions on the WRF model, the resolution of the model seriously affects the rainfall simulation, and there are large differences in the propagation of simulated metrics between model structures [21]. To analyze the effect of the model resolution, two different downscaling ratios (1:3 and 1:9) are employed. The results show that higher downscaling ratio leads to higher variability, which leads to a larger bias in the model simulation. The effect of small changes in the regional resolution on the simulation of the precipitation spatial pattern is greater than the impact on the predicted rainfall [22]. Finding the optimal set of physical parameterization schemes (and selecting appropriate model grid resolutions and LBCs) to simulate rainfall events and understanding the impact of different parameterization schemes on rainfall simulation in the Ili Region would be a valuable study.

This study used two reanalysis datasets and thirty-six members of a multiphysics ensemble for the WRF model simulation. The layout of this article is as follows: Section 2 provides a description of the reanalysis data used in the model, and the model configurations, sensitivity of physical parameterization schemes, and experimental setup are also discussed; the sensitivity of the WRF simulation to different LBCs and parameterization schemes is discussed in Section 3; and a summary of this research and proposed future study directions are included in Section 4.

## 2. Materials and Methods

### 2.1. Study Area and Observational Data

The Ili Region is located in the hinterland of Tianshan in Xinjiang between 80.1–84.5° E and 42.1–44.5° N (Figure 1). The study area is approximately 9460 km$^2$ and is the area with the most abundant precipitation in Xinjiang, with an annual precipitation between 200 and 550 mm, but the precipitation distribution in this area is uneven. To evaluate the performance of WRF model simulations, we used measurements of the hourly precipitation and hourly temperature from 131 stations, which consist of 121 telemetric stations and 10 climate stations that span across the Ili Region (Figure 1). The observation data were provided by the China Meteorology Administration (CMA).

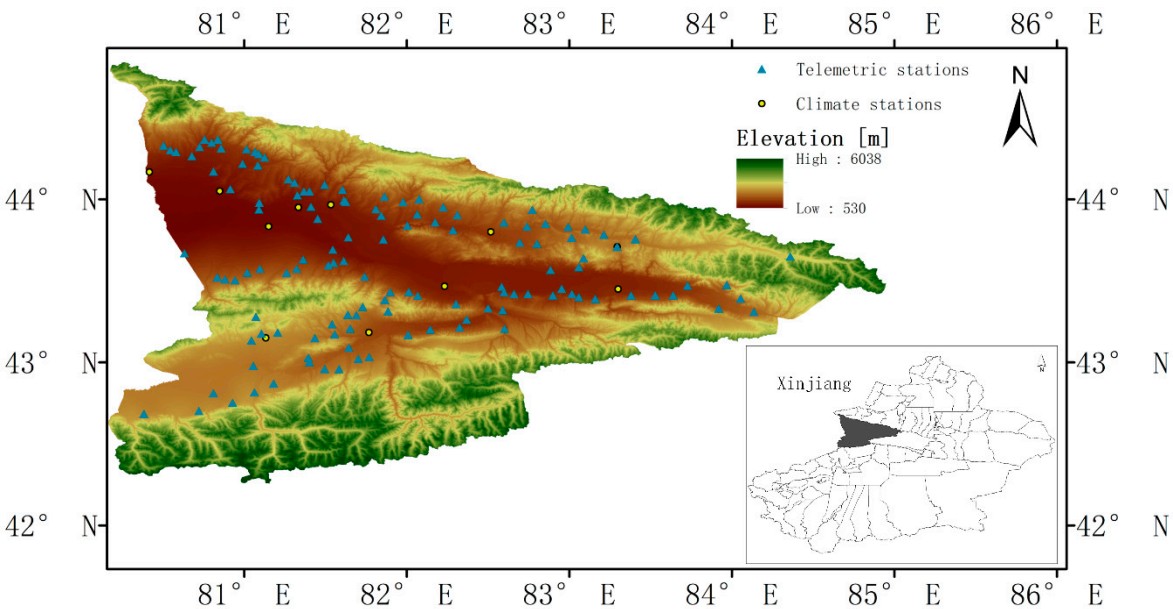

**Figure 1.** The study area (the points are telemetric stations, and the triangles are climate stations).

## 2.2. Model Configuration and Experimental Design

The WRF model configurations are shown in Table 1. The spatial setup consists of two domains, a parent domain of a 27-km resolution with $100 \times 120$ grid points, and a one-way nested 9-km resolution with $72 \times 69$ grid points. The time steps of the two domains are 180 seconds and 60 seconds respectively. This study focuses on high-resolution integration, so only the 9-km domain was analyzed. All the WRF simulations were initialized on 17 June 2013 and ended on 20 June 2013.

This study includes two different PBL schemes: the YSU scheme [23] and MYJ scheme [24]. Two different CU parameterization schemes are used: the Kain–Fritsch (KF) scheme [25] and the Grell–Devenyi (GD) scheme [26]. Three different microphysics parameterization (MP) schemes are used: the WRF Single Moment 6-class (WSM6) scheme [27], Thompson (THM) scheme [28], and Purdue Lin (Lin) scheme [29]. Four different longwave and shortwave radiation (RA) schemes are used: the Dudhia scheme [30], Rapid Radiative Transfer Model (RRTM) scheme [31], GFDL scheme [32], and New Goddard scheme [33]. A total of thirty-six members were produced for the simulation domain, as shown in Table 2.

Using the above settings, we performed two 3-day simulations spanning from 20130617 0:00 UTC to 20130620 0:00 UTC. The two experiments have identical settings with the exception of the initial and boundary conditions. One experiment (WRF-FNL experiment) used the FNL reanalysis data with a spatial resolution of $1° \times 1°$ and a temporal interval of 6 h. The other experiment (WRF-GFS experiment) used the GFS reanalysis data with a spatial resolution of $0.5° \times 0.5°$ and a temporal interval of 6 h.

**Table 1.** Model configurations.

| Model Options | Dataset or Value |
|---|---|
| Domains | 2 |
| Grid resolution (spacing) | 27:9 KM |
| Initial conditions | 1. Final Analysis (FNL) ($1° \times 1°$, 6 h); 2. Global Forecast System GFS ($0.5° \times 0.5°$, 6 h) |
| boundary layer (PBL) schemes | 1. Yonsei University (YSU); 2. the Mellor–Yamada–Janjic (MYJ) |
| Cumulus (CU) schemes | 1. Kain–Fritsch (KF); 2. Grell–Devenyi (GD) |
| Microphysics (MP)schemes | 1. the WRF Single Moment 6-class (WSM6); 2. Thompson (THM); 3. Purdue Lin (Lin) |
| Shortwave/Longwave radiation (RA)schemes | 1. Dudhia/Rapid Radiative Transfer Model (RRTM); 2. The Geophysical Fluid Dynamics Laboratory (GFDL)/The Geophysical Fluid Dynamics Laboratory (GFDL); 3. New Goddard/New Goddard |

**Table 2.** Ensemble design, physics options for PBL: YSU and MYJ; CU scheme: KF and GD; MP scheme: WSM6, THM, and Lin; RA schemes: Dudhia/RRTM, GFDL/GFDL, New Goddard/New Goddard.

| Member | PBL | CU | MP | RA |
|--------|-----|-----|------|-----|
| 1 | YSU | KF | WSM6 | Dudhia/RRTM |
| 2 | YSU | KF | WSM6 | GFDL/GFDL |
| 3 | YSU | KF | WSM6 | New Goddard/New Goddard |
| 4 | YSU | KF | THM | Dudhia/RRTM |
| 5 | YSU | KF | THM | GFDL/GFDL |
| 6 | YSU | KF | THM | New Goddard/New Goddard |
| 7 | YSU | KF | Lin | Dudhia/RRTM |
| 8 | YSU | KF | Lin | GFDL/GFDL |
| 9 | YSU | KF | Lin | New Goddard/New Goddard |
| 10 | YSU | GD | WSM6 | Dudhia/RRTM |
| 11 | YSU | GD | WSM6 | GFDL/GFDL |
| 12 | YSU | GD | WSM6 | New Goddard/New Goddard |
| 13 | YSU | GD | THM | Dudhia/RRTM |
| 14 | YSU | GD | THM | GFDL/GFDL |
| 15 | YSU | GD | THM | New Goddard/New Goddard |
| 16 | YSU | GD | Lin | Dudhia/RRTM |
| 17 | YSU | GD | Lin | GFDL/GFDL |
| 18 | YSU | GD | Lin | New Goddard/New Goddard |
| 19 | MYJ | KF | WSM6 | Dudhia/RRTM |
| 20 | MYJ | KF | WSM6 | GFDL/GFDL |
| 21 | MYJ | KF | WSM6 | New Goddard/New Goddard |
| 22 | MYJ | KF | THM | Dudhia/RRTM |
| 23 | MYJ | KF | THM | GFDL/GFDL |
| 24 | MYJ | KF | THM | New Goddard/New Goddard |
| 25 | MYJ | KF | Lin | Dudhia/RRTM |
| 26 | MYJ | KF | Lin | GFDL/GFDL |
| 27 | MYJ | KF | Lin | New Goddard/New Goddard |
| 28 | MYJ | GD | WSM6 | Dudhia/RRTM |
| 29 | MYJ | GD | WSM6 | GFDL/GFDL |
| 30 | MYJ | GD | WSM6 | New Goddard/New Goddard |
| 31 | MYJ | GD | THM | Dudhia/RRTM |
| 32 | MYJ | GD | THM | GFDL/GFDL |
| 33 | MYJ | GD | THM | New Goddard/New Goddard |
| 34 | MYJ | GD | Lin | Dudhia/RRTM |
| 35 | MYJ | GD | Lin | GFDL/GFDL |
| 36 | MYJ | GD | Lin | New Goddard/New Goddard |

*2.3. Evaluation Statistics*

The ability of the thirty-six members to simulate the precipitation and temperature of the two experiments were evaluated by the Bias (the mean bias), MAE (the mean absolute error), and RMSE (the root means square error):

$$\text{Bias} = \frac{1}{N} \sum_{i=1}^{N} M_i - O_i, \tag{1}$$

$$\text{MAE} = \frac{1}{N} \sum_{i=1}^{N} |(M_i - O_i)|, \tag{2}$$

$$\text{RMSE} = \sqrt{\frac{1}{N} \sum_{i=1}^{N} (M_i - O_i)}, \tag{3}$$

The spatial similarity at the grid cell level is measured with R (the pattern correlation coefficient) (Equation (4)) and the standard deviation ratio between the modeled value and the observation value (Equation (5)), and they can be reflected on Taylor diagrams (Section 3.2):

$$\text{R} = \frac{\sum_{i=1}^{N} (M_i - \overline{M})(O_i - \overline{O})}{\sqrt{\sum_{i=1}^{N} (M_i - \overline{M})^2} \sqrt{\sum_{i=1}^{N} (O_i - \overline{O})^2}}, \tag{4}$$

$$\sigma = \frac{\sqrt{\sum_{i=1}^{N} \left( M_i - \overline{M} \right)^2}}{\sqrt{\sum_{i=1}^{N} \left( O_i - \overline{O} \right)^2}}, \tag{5}$$

where $N$ is the total number of comparisons, $M$ is the simulations, $\overline{M}$ is average of simulations, and $O$ is the observations, $\overline{O}$ is average of observations.

The mean-metric is calculated to determine the overall performance of the best integrated members in the simulation of the precipitation and 2-m temperature, and the mean-metrics used for the ranking are the RMSE, MAE, and R score. Before sorting the collection members, we need to align the measurements: the RMSE and MAE scores were standardized by their respective maximum values, and the R score was inverted, so that a smaller value represents a better simulation effect. Now, the three indicators are all in the range of 0–1, and the simulation with the lowest average mean-metric is the best performing simulation.

## 3. Results

### 3.1. Verification of WRF Simulations

3.1.1. Climatological Spatial Pattern of Precipitation and 2-m Temperature

Figure 2 shows the performance of the precipitation simulation for the WRF-FNL experiment and WRF-GFS experiment. The average statistical indicators include the Bias (Equation (1)), MAE (Equation (2)), and R (Equation (4)). They were calculated for the modeled value and for the observations of the hourly rainfall of 131 stations that were then averaged.

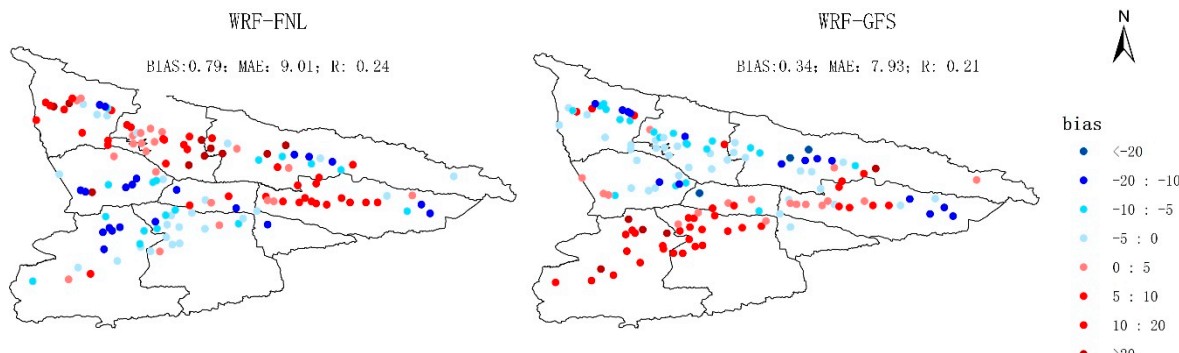

**Figure 2.** Geographical distribution of the precipitation simulation.

The average statistics of the WRF-FNL experiment and the WRF-GFS experiment show no significant difference. The mean of the WRF-FNL experiment is slightly worse. The Bias (0.79 mm) and MAE (9.01 mm) of the WRF-FNL experiment are larger than the Bias (0.34 mm) and MAE (7.93 mm) of the WRF-GFS experiment, and the correlation coefficients are 0.24 and 0.21, respectively. The R of the WRF simulations is not good due to the deviation in the parameterization scheme selection or the lack of a high horizontal grid spacing.

The main difference is that the geographical distribution of the WRF-FNL simulation precipitation generally decreased in the northeast and southwest and increased in the northwest and southeast. Furthermore, the WRF-GFS simulation precipitation generally decreased in the north and increased in the south. Therefore, WRF-FNL was generally similar to WRF-GFS for the precipitation simulation.

The simulation bias of the geographical distribution of the 2-m temperature is reflected in Figure 3. The average statistics of the WRF-FNL experiment and the WRF-GFS experiment show significant differences. The WRF-FNL experiment had a mean cold bias of 1.79 °C, while the WRF-GFS model predicted a mean warm bias of 1.81 °C in the region. The main difference is that the R and MAE of the WRF-GFS experiment (0.83 and 2.16, respectively) were better than those of the

WRF-FNL experiment (0.27 and 6.17, respectively). The geographical distribution of the WRF-FNL 2-m temperature simulation generally decreased in the north and increased in the southwest. However, the WRF-GFS 2-m temperature simulation generally increased over all the regions. In general, the 2-m temperature simulation of the WRF-GFS experiment was better than that of the WRF-FNL experiment.

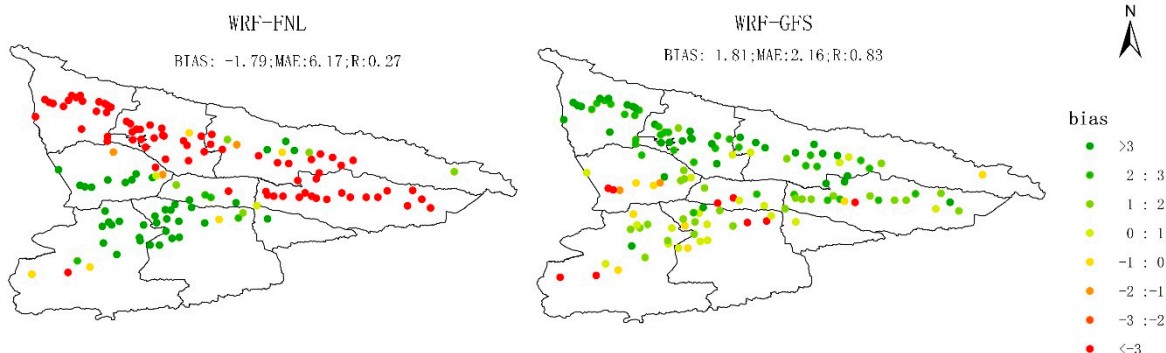

**Figure 3.** Geographical distribution of the 2-m temperature simulation.

The MAE (the absolute error) was calculated for the modeled value and for the observations of all stations. According to the MAE, the precipitation simulation results were divided into three categories: the first category was stations with the MAE value greater than 20 mm; The second type is stations with the MAE between 10–20 mm; The third type is stations with the MAE less than 10 mm. The 2-m temperature simulation results also can be divided into three categories: first, there was stations with the MAE value greater than 4 °C; The second type of stations with the MAE value between 2–4 °C; The third kind of stations with the MAE is less than 2 °C.

Table 3 shows the performance of the spatial simulation for the WRF-FNL experiment and WRF-GFS experiment. For the simulation of precipitation, both the WRF-FNL and WRF-GFS simulations were found to perform well. The GFS experiment did a little better because it had the MAE of less than 10 mm for 93 stations (the mean MAE was 4.63 mm for 93 stations), and there were 30 stations with MAE between 10 and 20 mm (of which the 30 stations had the mean MAE of 14.06 mm), and only 8 stations had an MAE greater than 20 mm (the mean MAE of these 8 stations was 23.34 mm). In the simulation results of the WRF-FNL experiment, the number of stations with the MAE of less than 10 mm (81 stations) was 12 fewer than the WRF-GFS experiment, and the number of stations with the MAE of more than 20 mm (10 stations) was 2 more than the WRF-GFS experiment. For the simulation of the 2-m temperature, the simulation of the WRF-GFS experiment (most of the stations (121 stations) were less than 4 °C) was generally better than that of the WRF-FNL experiment (most of the stations (95 stations) were greater than 4 °C). Therefore, the WRF-GFS experiment has a better simulation effect from the overall simulation results.

**Table 3.** The results of spatial simulation of precipitation and 2-m temperature.

| Variable | Absolute Error | WRF-GFS Experiment | | WRF-FNL Experiment | |
|---|---|---|---|---|---|
| | | **Mean** | **Number of Stations** | **Mean** | **Number of Stations** |
| precipitation | >20 mm | 23.34 mm | 8 | 28.16 mm | 10 |
| | 10–20 mm | 14.06 mm | 30 | 13.81 mm | 40 |
| | <10 mm | 4.63 mm | 93 | 4.28 mm | 81 |
| 2-m temperature | >4 °C | 4.63 °C | 10 | 7.67 °C | 95 |
| | 2–4 °C | 2.89 °C | 58 | 2.98 °C | 21 |
| | <2 °C | 1.09 °C | 63 | 1.1 °C | 15 |

### 3.1.2. Temporal Characteristics of Rainfall Events

The WRF-FNL simulation results are shown in Figure 4. For the WRF-FNL simulation, the main rainfall event started at 20130617 10:00 UTC and ended at 20130617 18:00 UTC, and between 20130618

12:00 UTC and 20130619 12:00 UTC, with most simulations indicating 6 h earlier for the onset and termination of the event. Most of the thirty-six members have the tendency to overestimate the accumulated precipitation, particularly at 20130618 12:00 UTC to 20130619 0:00 UTC. The bias between the thirty-six members of the WRF-FNL experiment and the observation data is reflected in Figure 4b, indicating a range between −0.67 and 1.33 mm. The partial view highlights the two sets of simulations (N1–9 and N19–27) that display a significant overestimation of the peak precipitation; these correspond to the simulation of the KF scheme. The set of thirty-six members can obtain a smaller error for peak precipitation, and those simulations (N10–18) are the YSU scheme in combination with the GD scheme. In general, the GD scheme is slightly better than the KF scheme. The red line in Figure 4a is the mean of the hourly precipitation simulations of the thirty-six members, displaying a temporal evolution of the event very similar to that of the observations. However, none of the physics scheme combinations stand out as the best performer.

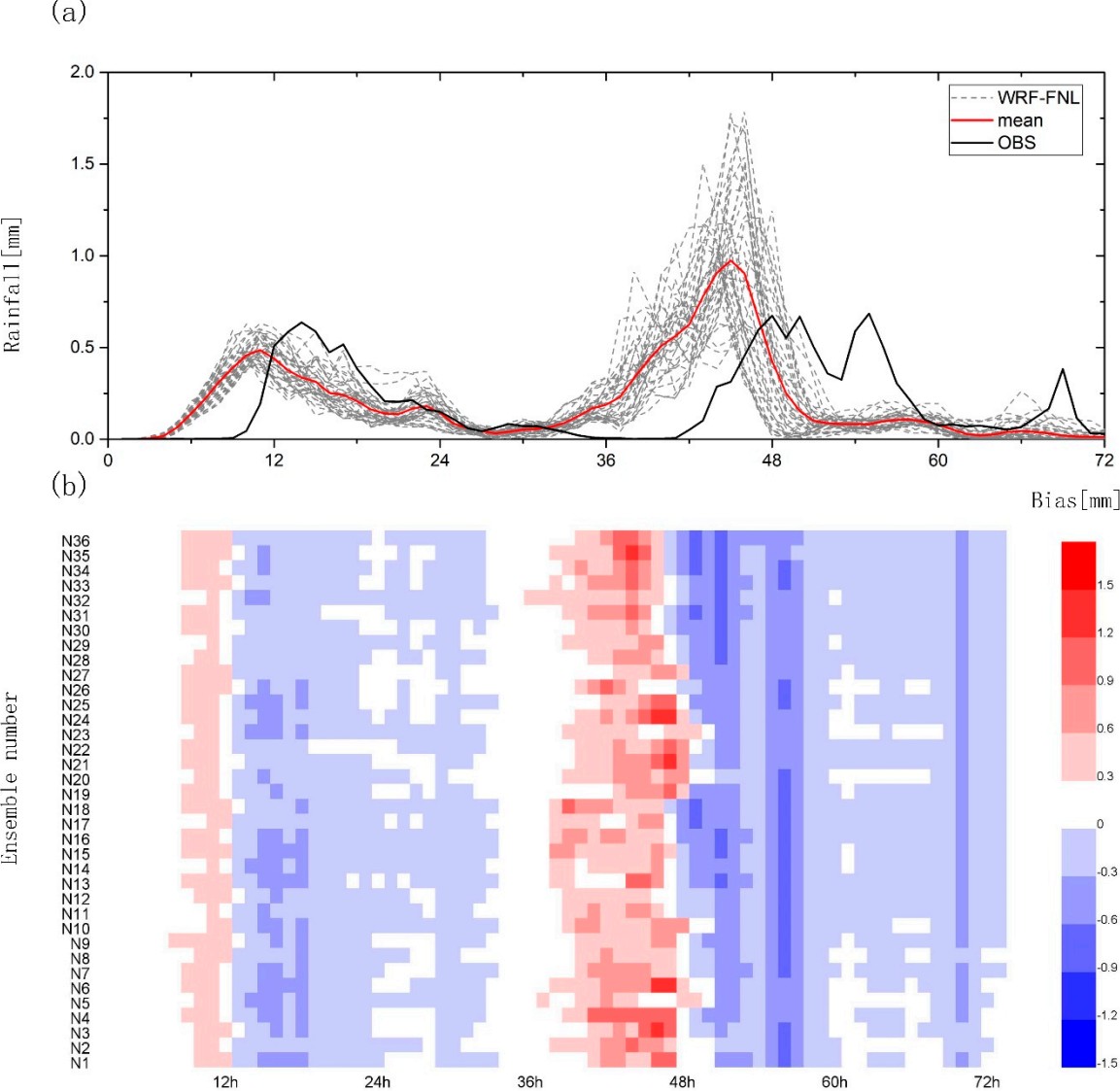

**Figure 4.** (**a**) is the mean hourly precipitation (mm) of the WRF-FNL experiment for the thirty-six members; (**b**) is the bias (mm) for each hour of the thirty-six members.

The WRF-GFS simulation results are shown in Figure 5. All of the thirty-six members show good performance in modeling the total accumulated precipitation. The bias between the WRF-GFS experiment members and the precipitation observations is reflected in Figure 5b, indicating a range

between −0.48 and 1.45 mm. The capability of the WRF-GFS experiment is comparable to that of the WRF-FNL experiment. However, many of the thirty-six members have the tendency to overestimate a secondary event in the middle of the period. The red line in Figure 5a is the mean of the hourly precipitation simulations of the thirty-six members, displaying a very similar temporal evolution. Like the WRF-FNL experiment, the YSU scheme and the GD scheme can also obtain higher simulation accuracies. No preference for a particular MP scheme or RA scheme was found.

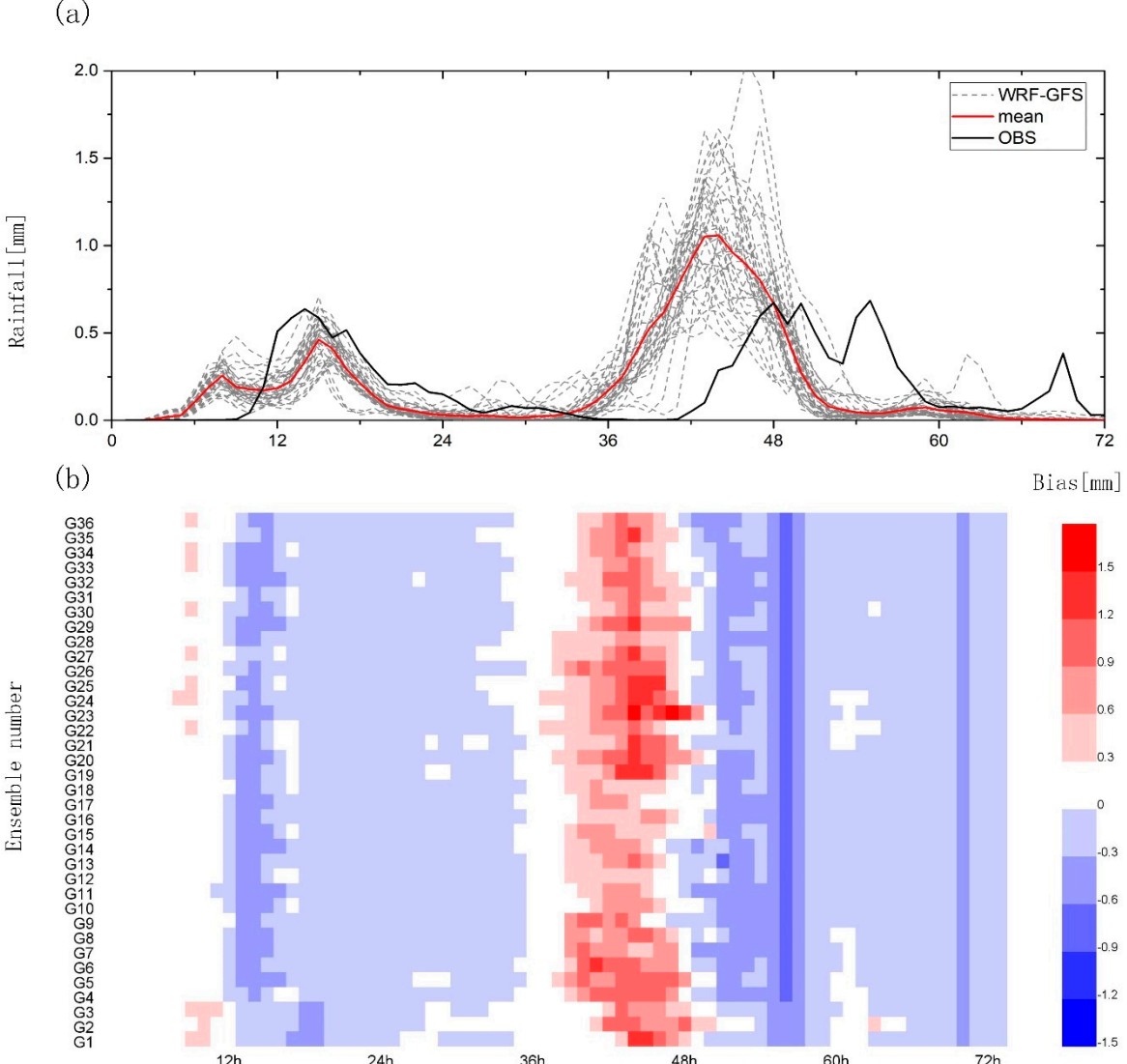

**Figure 5.** (**a**) is the mean hourly precipitation (mm) of the WRF-GFS experiment for the thirty-six members; (**b**) is the bias (mm) for each hour of the thirty-six members.

Both the WRF-FNL simulation and the WRF-GFS simulation can capture the general temporal pattern of the rainfall event, characterized by peak rainfalls at 20130617 1200 UTC and 20130619 0:00 UTC (Figure 4). The temporal patterns of rainfall from the WRF-FNL simulation are a better match to the observed values than those from the WRF-GFS experiment. However, both simulations overestimated the precipitation at 20130619 0:00 UTC, although the WRF-GFS simulation overestimated the precipitation by more than the WRF-FNL simulation.

In Figure 6a, the black columns are the bias of the cumulative precipitation error of the thirty-six members in the WRF-FNL experiment and the red columns are the same for the WRF-GFS experiment. The WRF-FNL simulation indicates a range between −2.99 and 4.52 mm and the WRF-GFS simulation

indicates a larger range between −4.78 and 6.24 mm. Large positive biases are shown for both the WRF-FNL simulation and the WRF-GFS simulation when using the KF scheme, whereas the GD scheme is associated with slightly smaller biases in G10–18 and G28–36. The bias values of the hourly mean for the 2-m temperature are shown in Figure 6b. The WRF-FNL simulation overestimates the 2-m temperature to some degree in almost all case studies, indicating a range between −2.41 and −1.20 °C. The WRF-GFS simulation tended to underestimate the 2-m temperature in all case studies, indicating a range between 1.25 and 2.62 °C. None of the physics scheme combinations stand out as the best performer.

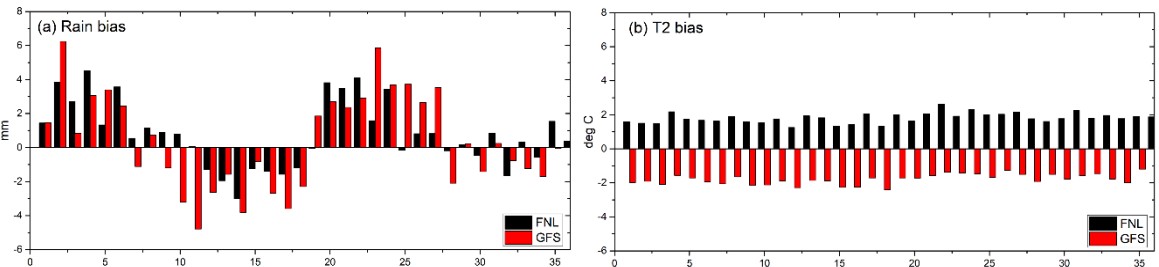

**Figure 6.** Bias in (**a**) rain and (**b**) T2 for each ensemble member.

### 3.2. Impact of Different Parameterization Schemes

The Taylor diagram (Figure 7) shows the σ (the standard deviation ratio, calculated by Equation (5)) and R between the thirty-six members and the observations. (a) and (b) provide the standard to evaluate the capability of the different parameter schemes for the WRF-FNL experiment, and (c) and (d) show the same for the WRF-GFS experiment. The rainfall simulations of the parameterization schemes are reflected in (a) and (c), and the 2-m temperature simulations are shown in (b) and (d).

Small differences are observed in the performances of the various parameterizations. However, differences between the performances of the MP schemes can be found in several simulations. This finding is shown in Figure 7a, where Thompson (blue) performs better than WSM6 (red) or Lin (black). For the rest of the simulation, these schemes perform similarly. The difference in the PBL schemes is large (second only to MP schemes), but the simulation values of each PBL scheme are not very different. The YSU scheme has marginally better performance (sometimes) in the WRF-FNL experiment for the rainfall simulations, and the MYJ scheme performed the better WRF-GFS experiment for the 2-m temperature simulations. Therefore, the MYJ scheme outperformed the YSU scheme in these comparisons. The RA and CU schemes both tended to have only small differences that are generally not large enough to differentiate the two sets of simulations. Figure 7c shows a slight preference for the GD scheme. Figure 7b shows small differences in the simulation quality produced.

A box diagram (shown in Figures 8 and 9) can reflect a whole process for selecting the best member. After calculating the mean metric ranges of each physics parameterization scheme, the scheme with the smallest mean metric can be then chosen. If one option of the physical scheme performs better (or worse) than the other options and the value of the average metric range is smaller (or larger), then this option is considered the preferred (or rejected) physical scheme. In the first step of this method, the mean-metric ranges for all physics schemes are calculated, and then the best (or worst) physical scheme was selected (or eliminated). In the next step, the mean metric ranges are calculated again using the remaining members. This method was repeated until the best performing member was selected, and each step provides the mean-metric of the chosen member.

Figure 8 shows box plots for the stepwise approaches to determine the best physical scheme for the precipitation simulation, where the first row (Figure 8a–d) of each box plot shows the mean metric of precipitation of all 36 ensemble members. When comparing ensemble members using the mean metric of precipitation, both the minimum (0.53) and maximum (0.68) values for the GD CU scheme

are smaller than the minimum (0.56) and maximum (0.90) values of the KF CU scheme. The mean metric ranges of all the remaining physical parameterizations are not significantly different. Hence, the second row (Figure 8e–g) of each box plot shows only models with the GD scheme. The third row (Figure 8h,i) includes models with the GD CU scheme and the MYJ PBL scheme because the mean metric ranges (0.53–0.62) of the MYJ PBL scheme are smaller than those (0.59–0.68) of the YSU PBL scheme. The choice to remove the GFDL RA scheme is shown in Figure 8j, as its mean metric ranges are only larger than those of the other RA schemes. This process results in six ensemble members (28, 30, 31, 33, 34, and 36) that cannot be robustly differentiated based on their parameterizations. In summary, the best scheme is member 33, but it cannot be distinguished from the other members (28, 30, 31, 34, and 36) according to its parameterizations.

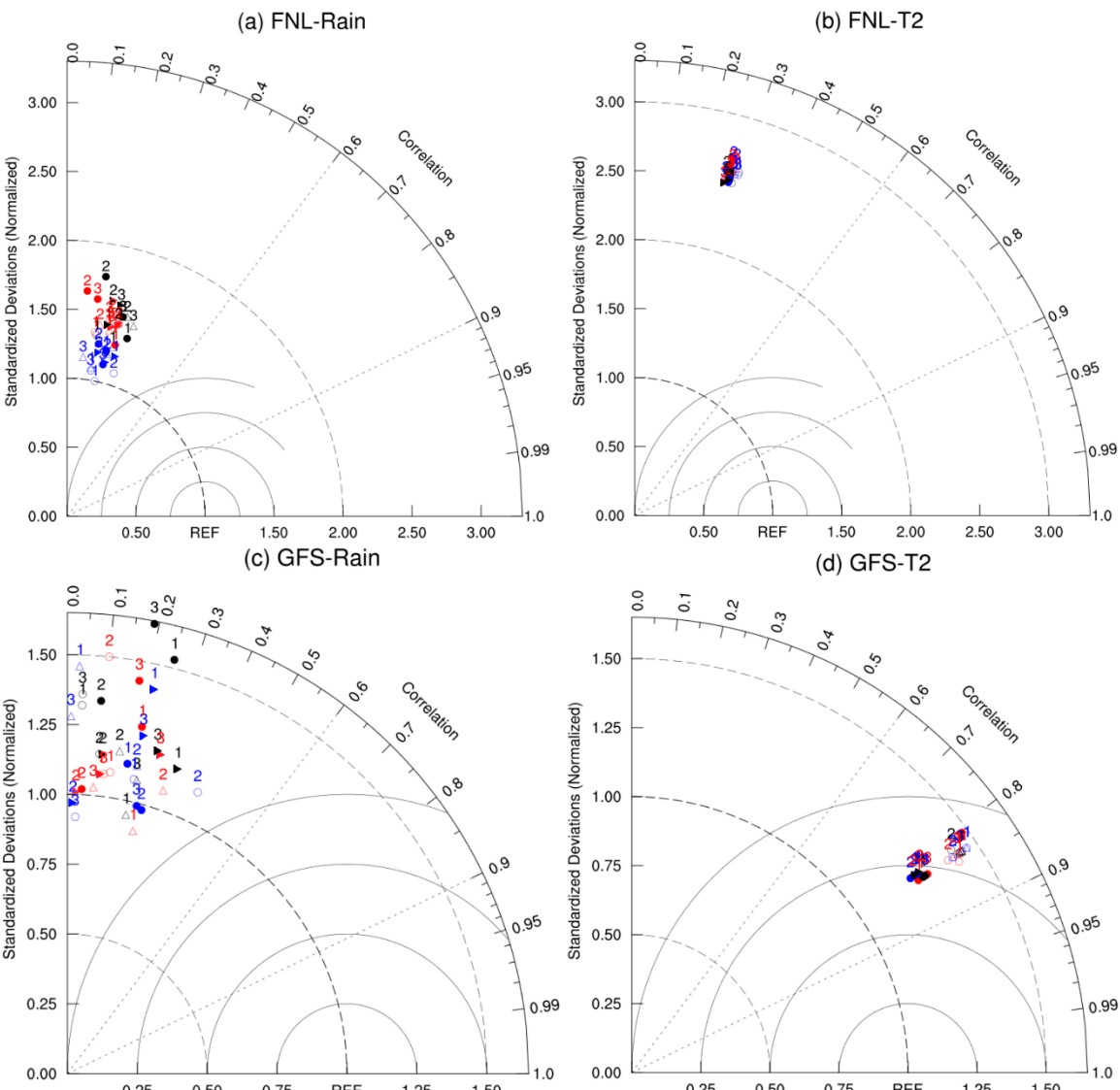

**Figure 7.** Taylor diagrams for different parameterization schemes in two experiments. Colors are different MP schemes: WSM6 (red), Thompson (blue), and Lin (black); filled or hollow mean different PBL schemes: YSU (hollow) and MYJ (filled); shapes mean different CU schemes: KF (circle) and GD (triangle); numbers represent the RA schemes: RRTM (1), GFDL (2), and New Goddard (3).

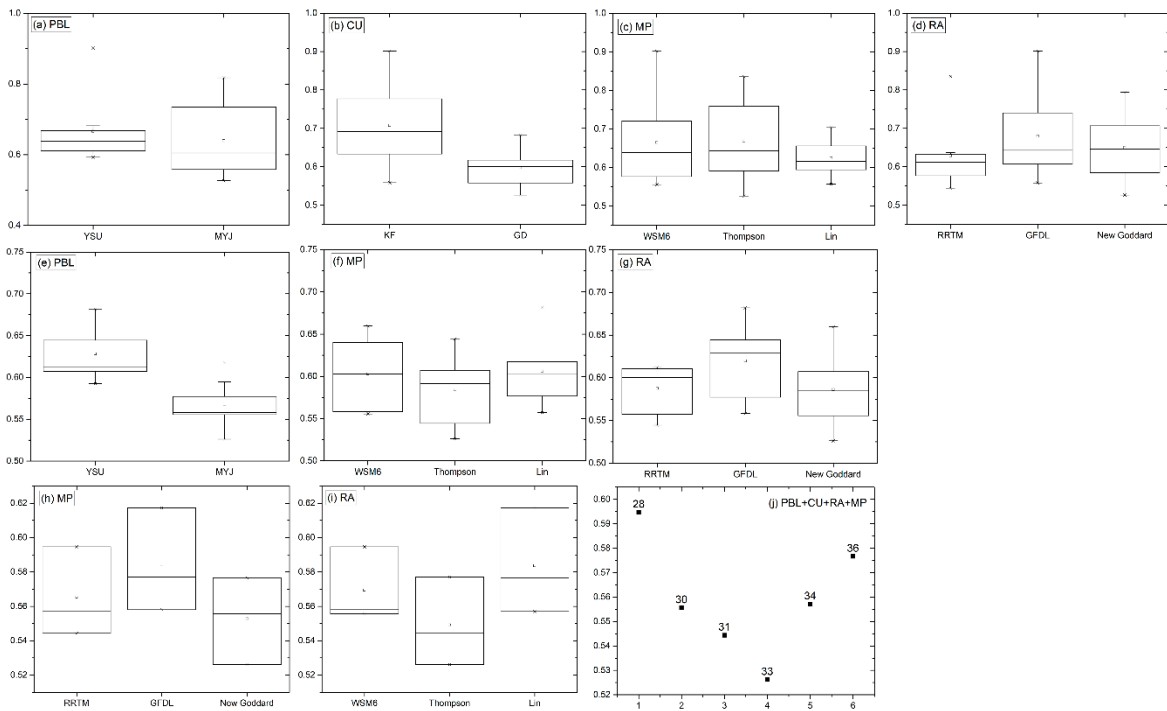

**Figure 8.** Box plots of the mean-metric of precipitation split by physical parameterization. In (**a–d**), all 36 ensemble members are shown. In (**e–g**), the plots include only models with the GD scheme. In (**h,i**), the plots include models with the GD scheme and the MYJ scheme. In (**j**), the plot includes models with the GD scheme and the MYJ scheme but excludes the GFDL scheme.

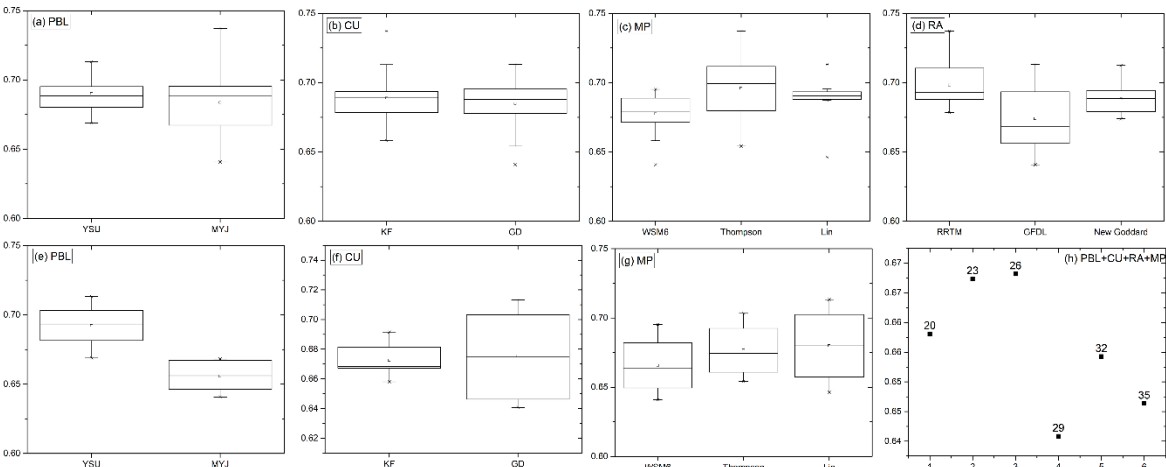

**Figure 9.** Box plots of the mean-metric of the 2-m temperature split by physical parameterization. In (**a–d**), all 36 ensemble members are shown. In (**e–g**), the plots include only models with the GFDL scheme. In (**h**), the plot includes models with the GFDL scheme and the MYJ scheme.

Figure 9 shows box plots for the stepwise approaches to determine the best physical scheme for the 2-m temperature simulation. In Figure 9a–d, thirty-six ensemble members are shown. The GFDL RA scheme with the smallest mean metric is the simulation with the best performance; thus, the GFDL is selected, and Figure 9e–g includes only models with the GFDL RA scheme. The most robust choice is the MYJ PBL scheme, as its ensemble member range (0.64–0.67) is smaller than that of the YSU PBL scheme (minimum is 0.67); thus, these ensemble members (2, 5, 8, 11, 14, and 17) are removed. In the final step (Figure 9h), the plot includes models with the GFDL RA scheme and the MYJ PBL

scheme. This process results in six ensemble members (20, 23, 26, 29, 32, and 35) that cannot be robustly differentiated based on their parameterizations, although the best scheme seems to be member 29.

## 4. Conclusions

Detailed climate scenarios are not available for the Ili Region because of the lack of observations and the complex topography in the area. The global reanalysis data is too rough to accurately express the hydrological climate of the Ili Region. To date, no dynamic downscaling studies have been conducted to record high-resolution climates in the region. The present study using the WRF model with 9-km high-resolution is the first to dynamically downscale the FNL and GFS datasets in the Ili Region. Sensitivity studies were performed using the CU, PBL, RA, and MP schemes.

Two reanalysis datasets (including the FNL and GFS datasets) are provided to verify the simulation effect of the WRF model. The results showed that the simulations of the WRF model have some discrepancies from the observation data. For the precipitation simulation, both the WRF-FNL simulation (mean bias of 0.79 mm) and the WRF-GFS simulation (mean bias of 0.31 mm) have weak positive biases. However, the geographical distribution of the WRF-FNL simulation precipitation shows a generally negative bias in the northeast and southwest and a positive bias in the northwest and southeast. The WRF-GFS simulation precipitation generally decreased in the north and increased in the south. For the simulation of the 2-m temperature, the WRF-FNL simulation predicted a mean cold bias of 1.79 °C, and the WRF-GFS model predicted a mean warm bias of 1.81 °C. The geographical distribution of the WRF-FNL simulation of the 2-m temperature generally showed decreases in the north and increases in the southwest. However, the WRF-GFS simulation of the 2-m temperature generally increased over all regions.

Some results were collected on the performances of the physical parameterization members. The differences between the physical members of the simulated 2-m temperature are much smaller than those of the precipitation simulation. For the WRF model parameterization study, none of the physical members have the best performance in any situation, although different analyses reveal the preferences of some physics schemes. In terms of the time process simulation, consistently identifying well-performing physics combinations across all case studies is difficult. The overall bias analysis shows that the rainfall simulation is more sensitive to CU schemes, and the bias of GD schemes is generally lower. For the 2-m temperature, no physics scheme combination stands out as being the best performer. A comparison of the MAE, RSME, and R can reveal a preference for the THM schemes, a slight preference for the MYJ schemes, and unclear preferences for the RA and CU schemes.

By calculating the mean-metric range of each physics parameterization scheme, for the precipitation simulation, ensemble member 33 performs the best. For the simulation of the 2-m temperature, the best-performing member is 29. In the Ili Region, the MYJ and GD schemes were recommended to simulate the precipitation, and the GFDL RA scheme and the MYJ PBL scheme are recommended for the simulation of the 2-m temperature.

One of the limiting factors for this type of research is the lack of observations for proper verification in the Ili Region. In addition, this study used only three days of simulation time. Further work is ongoing to repeat this analysis for more events to more fully assess all rainfall patterns. The simulation result will then be further analyzed to determine the impact of different parameterization schemes on the performance of the WRF model, which is beyond the preliminary results of this paper. The combination of better performing parameterization schemes found here will be used in multidecadal simulations to further assess WRF's capability to simulate the precipitation and 2-m temperature.

**Author Contributions:** Y.Z. and Z.M. conceived the study, developed the methodology; Y.Z. performed model simulation with the collected data, and analyzed and organized the model results into figures and tables; Z.M. contributed to manuscript revision.

**Acknowledgments:** This research was supported by the National Natural Science Foundation of China (Grant Nos. 51469034, 51209181 and 51569031), The key discipline research project of water conservancy engineering

of xinjiang agricultural university (Grant Nos.SLXK2018-02). The Natural Science Foundation of xinjiang uygur autonomous region (Grant Nos.2018D01A16).

**Conflicts of Interest:** The authors declare no conflict of interest.

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
