# Peer review of "Impact of Different Reanalysis Data and Parameterization Schemes on WRF Dynamic Downscaling in the Ili Region"

_water, doi:10.3390/w10121729_

Reviewer 1 Report

This manuscript describes an effort to estimate various combinations of the WRF parameters using Final Analysis (FNL) and Global Forecast System (GFS) reanalysis data. This paper did a decent comparison among these combinations. I think the work is interesting for the general audience of the journal. However, in its present form, the contribution of the paper falls short of the standard required for publication.

First paragraph of Introduction, In general, the "climate change" refers to a change in average weather conditions, when that change lasts for an extended period of time (i.e., decades to millions of years). That is, the time variation of weather might occur around longer-term average conditions. However, the length of only 3-day simulations was used to treat the precipitation issue, not addressed the influence regarding "climate change". I suggest "climate change" should be removed, unless the authors provided some analysis regarding how the "climate change" affect the rainfall and temperature in the Ili Region.

This study used only three days of simulation time, although the authors declared the lack of observations for proper verification in the Ili Region. The best scheme combinations are not quite convincing. That is, when another weather events and dates in the Ili Region, the best scheme combinations might be different. A more weather cases should be provided to verify these "best combinations".

The paper does not contain a clear contribution. While the various combination of WRF schemes to estimation of rainfall and temperature are interesting, I was unable to discern from the results presented what we were actually supposed to learn from the exercise. Nothing was revealed about the physical or climatic side of scheme combinations. For example, why (1) the MYJ and GD schemes were recommended to simulate the precipitation, and (2) GFDL RA scheme and the MYJ PBL scheme are recommended for the simulation temperature in the Ili Region. Suggested new section can be added before Conclusions.

Minor suggestions:

1) page 5, Equations (4) and (5), M and O should be average of sim and obs. Please change these symbols, because they confuse with the following sentence "where N is the total number of comparisons, M is the simulations, and O is the observations" .

Author Response

Dear reviewer

This refers to your previous mail dated Nov 7th 2018 on the decision for “Impact of different reanalysis data and parameterization schemes on WRF dynamical downscaling in the Ili Region”.

Thank you for your valuable comments on this article. According to your suggestions, we make necessary corrections and responses.

Response to Reviewer 1 Comments

Point 1:  First paragraph of Introduction, In general, the "climate change" refers to a change in average weather conditions, when that change lasts for an extended period of time (i.e., decades to millions of years). That is, the time variation of weather might occur around longer-term average conditions. However, the length of only 3-day simulations was used to treat the precipitation issue, not addressed the influence regarding "climate change". I suggest "climate change" should be removed, unless the authors provided some analysis regarding how the "climate change" affect the rainfall and temperature in the Ili Region.

Response 1: Many thanks for your suggestions. We used the 3-day simulations to treat the precipitation issue and have removed "climate change".

Point 2: This study used only three days of simulation time, although the authors declared the lack of observations for proper verification in the Ili Region. The best scheme combinations are not quite convincing. That is, when another weather events and dates in the Ili Region, the best scheme combinations might be different. A more weather cases should be provided to verify these "best combinations".

Response 2: To validate the applicability of the WRF model in this study area, we simulated for specific rainfall events to find the most appropriate combination scheme. At present, the data collected in the study area is only 2012-2014. The precipitation sequence is selected as the strong precipitation event with the complete data and the most available stations in the study area. Therefore, this selection is based on the availability of observations.  Compared with other studies, we found that the WRF model is not suitable for small rainfall events. In order to better serve the runoffflood forecasting and flood mitigation, we chose this strong precipitation event to find the most suitable combination for large rainfall events.

Point 3: The paper does not contain a clear contribution. While the various combination of WRF schemes to estimation of rainfall and temperature are interesting, I was unable to discern from the results presented what we were actually supposed to learn from the exercise. Nothing was revealed about the physical or climatic side of scheme combinations. For example, why (1) the MYJ and GD schemes were recommended to simulate the precipitation, and (2) GFDL RA scheme and the MYJ PBL scheme are recommended for the simulation temperature in the Ili Region. Suggested new section can be added before Conclusions.

Response 3: By calculating the mean-metric of each physical scheme in WRF-FNL experiment and WRF-GFS experiment for the simulation of precipitation and 2-m temperature, which is between 0 and 1. The smaller the mean-metric, the better the simulation effect.

For the simulation of precipitation, we calculated the average of the mean-metric for WRF-FNL experiment and WRF-GFS experiment. And we found that in the PBL scheme, the MYJ scheme (the mean-metric between 0.53 and 0.62) has smaller metrics compared to other PBL schemes; in the CU scheme, the GD scheme (the mean-metric between 0.53 and 0.68) has smaller metrics compared to other CU schemes. Therefore, (1) the MYJ and GD schemes were recommended to simulate the precipitation. For the simulation of 2-m temperature, we found that in the PBL scheme, the MYJ scheme (the mean-metric between 0.64 and 0.67) has smaller metrics compared to other PBL schemes; in the RA scheme, the GFDL scheme (the mean-metric between 0.64 and 0.71) has smaller metrics compared to other RA schemes. Therefore, (2) GFDL RA scheme and the MYJ PBL scheme are recommended for the simulation temperature in the Ili Region.

Minor comments: page 5, Equations (4) and (5), M and O should be average of sim and obs. Please change these symbols, because they confuse with the following sentence "where N is the total number of comparisons, M is the simulations, and O is the observations"

Response: Thank you for your advice. We have corrected the equations (4) and (5), and changed those symbols to represent the variable in the equations.

Dear reviewer, we have done a lot of efforts to make our manuscript acceptable. Our revised manuscript is ready for submission. We are waiting desperately for your answer and we hope it will be positive.

Yours sincerely,

Yulin Zhou

The first author

Zhenxia MU

Reviewer 2 Report

The article investigates the sensitivity of the WRF model due to different forcing data and due to a large number of combinations of parameterization schemes of physical processes. The study focuses on northwestern China and only over a specific summer event. Although the study is useful for the improvement of modeling effort in the region, my recommendation is to reconsider after a major revision. More specific points follow.

Major comments

- The simulation period is very short to draw safe conclusions (only one 3-day summer event). I would test at least one more event during a different period. Please highlight in the title (or at least in the abstract) that the study is only based on a single event.

- FNL and GFS data are not independent in order to draw general conclusions on the sensitivity because of the forcing data. I would either additionally test an independent reanalysis dataset (e.g. ERA-Interim) or focus only on the sensitivity to physics options for the FNL-driven runs.

- The authors should explain why they selected to downscale the specific event. Was this a strong precipitation event? Did it have a great impact? Was this selection based on the availability of observations?  

- In the discussion of Figure 2 you use the mean bias and correlation from all stations. This might not be safe since I can see that there are stations that indicate a strong positive bias, while there are also stations indicating a strong negative bias, therefore the mean values smooth the results. Same applies for the discussion of Figure 3. I recommend to use the absolute values and then average. Also, the caption of these two figures should change since only biases are presented.

- It is not clear if results are presented as averages for all stations. This should be clarified. Again, the averaging over a large region could smooth results and affect your conclusions. I recommend the grouping of stations that have similar characteristics and perform a separate analysis for sub-regions.

- Figures 4 and 5: the color scale should be centered at the value of 0 (e.g. red colors positive biases, blue colors negative biases or vice versa). They should have the same range (e.g. from -1.5 to 1.5 mm) and the same for both figures.  

- In section 3.2 and Figure 7c I cannot see a clear distinction for the MP selections as you mention in the text. This is evident only in Figure 7a.

- In page 10, the two paragraphs that describe the mean metric and your ranking process should be moved in the methods section.

- I recommend a refinement of your ranking methodology and the effort to find the “best” configurations towards more objective criteria. In the boxplots there is an overlap of the performance for some schemes. For example, some “good” KF simulations for precipitation are more skillful than the “bad” GD, however these are excluded. Same applies for other steps. This is also evident from the Taylor plots of Figure 7 where some “skillful” configurations (e.g. 5 and 10 for GFS rain are not included in your suggested ones. You should also clarify which runs were used for the boxplots (GFS? FNL? Or both?).

Minor comments

- Define MP in the introduction.

- Correct ERI-40 to ERA-40

- In equation (2) instead of MAE you define RMSE. Please correct.

- In the first paragraph of the results section you refer to equation (3) for R. I think the correct reference for R is equation (4). In the same section in the text you mention a mean Bias of 0.31 mm, while in Figure 2 this bias is 0.34. Please check again.

- I would skip the first four rows of the conclusions section.

Author Response

Dear reviewer

This refers to your previous mail dated Nov 7th 2018 on the decision for “Impact of different reanalysis data and parameterization schemes on WRF dynamical downscaling in the Ili Region”.

Thank you for your valuable comments on this article. According to your suggestions, we make necessary corrections and responses.

Point 1:The simulation period is very short to draw safe conclusions (only one 3-day summer event). I would test at least one more event during a different period. Please highlight in the title (or at least in the abstract) that the study is only based on a single event.

Response 1: Many thanks for your suggestions. We have supplemented these contents in the abstract.

Point 2: FNL and GFS data are not independent in order to draw general conclusions on the sensitivity because of the forcing data. I would either additionally test an independent reanalysis dataset (e.g. ERA-Interim) or focus only on the sensitivity to physics options for the FNL-driven runs. 

Response 2: Due to the complexity of the topography of the study area and the multi-source of water vapor sources, the complexity of the law of precipitation changes. Considering that the WRF model has not been studied in the Ili region, the FNL and GFS reanalysis data that widely used in the WRF model were selected to explore the applicability of WRF in this area and the best combination scheme. There are many available reanalyzed data, and the applicability of different reanalyzed data to different research areas are quite different. We will conduct a variety of reanalysis-data driven WRF models in the later stage.

Point 3:The authors should explain why they selected to downscale the specific event. Was this a strong precipitation event? Did it have a great impact? Was this selection based on the availability of observations? 

Response 3: We selected to downscale the specific event for the following reasons:(1) The research area started to build telemetry stations successively from 2009, and the construction of the stations reached a certain scale after 2012, and the operation was normal. During this time we only collected data from the telemetry station from 2012 to 2014. Therefore, this selection was based on the availability of observations; (2) This was a strong precipitation event, and it had complete observational data. Therefore, we decided to downscale this specific event.

Point 4:In the discussion of Figure 2 you use the mean bias and correlation from all stations. This might not be safe since I can see that there are stations that indicate a strong positive bias, while there are also stations indicating a strong negative bias, therefore the mean values smooth the results. Same applies for the discussion of Figure 3. I recommend to use the absolute values and then average. Also, the caption of these two figures should change since only biases are presented. 

Response 4: we are very sorry not to check equation of MAE (the mean absolute error), and the MAE covers what you mean by “the absolute values and then average”. In the discussion of Figure 2 and Figure 3, the MAE have been compared between two groups of experiments, but we did not display the value in the diagram. We have corrected equation (2) and put the MAE in the figure2 and Figure 3.

Point 5: It is not clear if results are presented as averages for all stations. This should be clarified. Again, the averaging over a large region could smooth results and affect your conclusions. I recommend the grouping of stations that have similar characteristics and perform a separate analysis for sub-regions.

Response 5:  

We should clarify that the results are presented as averages for all stations. According to your suggestions, the new section and table are added to analysis the grouping of stations that have similar characteristics.

The MAE (absolute error) was calculated for the modeled value and for the observations of all stations. According to the MAE, the precipitation simulation results were divided into three categories: the first category was stations with the absolute error value greater than 20mm; The second type was stations with absolute error between 10-20mm; The third type is stations with absolute error less than 10mm. The 2-m temperature simulation results also can be divided into three categories: first, there was stations with the error absolute value greater than 4°C; The second type of stations assumed the error absolute value between 2-4°C; The third kind of stations with the absolute error was less than 2°C.

        Table 3 shows the performance of the spatial simulation for the WRF-FNL experiment and WRF-GFS experiment. For the simulation of precipitation, both the WRF-FNL and WRF-GFS simulations were found to perform well. The GFS experiment did better because it had the MAE of less than 10mm for 93 stations (the mean MAE was 4.63mm for 93 stations), and there was 30 stations with the MAE between 10 and 20mm (of which the 30 stations had the mean MAE of 14.06mm),and the MAE of only 8 stations was greater than 20mm (the mean MAE of these 8 stations was 23.34mm). In the simulation results of WRF-FNL experiments, the number of stations with the MAE of less than 10 mm (81 stations) was 12 fewer than the WRF-GFS experiment, and the number of stations with the MAE of more than 20 mm (10 stations) was 2 more than the WRF-GFS experiment. For the simulation of the 2-m temperature, the simulation of the WRF-GFS experiment (most of the 121 stations were less than 4°C) was generally better than that of the WRF-FNL experiment (most of the 95 stations were greater than 4°C). Therefore, the WRF-GFS experiment has a better simulation effect from the overall simulation results.

                 Table 3. The results of spatial simulation of precipitation and 2-m temperature

absolute errorWRF-GFS experimentWRF-FNL experiment

MeanNumber of stationsMeanNumber of stations
precipitation>20mm23.34 mm828.16 mm10

10-20mm14.06 mm3013.81 mm40

<10mm< span="">4.63 mm934.28 mm81
temperature>4°C4.63°C107.67°C95

2-4°C2.89°C582.98°C21

<2°c< span="">1.09°C631.1°C15

Point 6: Figures 4 and 5: the color scale should be centered at the value of 0 (e.g. red colors positive biases, blue colors negative biases or vice versa). They should have the same range (e.g. from -1.5 to 1.5 mm) and the same for both figures.  

Response 6: According to your suggestion, we have corrected Figures 4 and 5 to the same range (from -1.5 to 1.5 mm).

Point 7: In section 3.2 and Figure 7c I cannot see a clear distinction for the MP selections as you mention in the text. This is evident only in Figure 7a.

Response 7: This is evident only in Figure 7a, we have corrected it.

Point 8: In page 10, the two paragraphs that describe the mean metric and your ranking process should be moved in the methods section.

Response 8: The two paragraphs that describe the mean metric and your ranking process have been moved in the methods section.

Point 9: I recommend a refinement of your ranking methodology and the effort to find the “best” configurations towards more objective criteria. In the boxplots there is an overlap of the performance for some schemes. For example, some “good” KF simulations for precipitation are more skillful than the “bad” GD, however these are excluded. Same applies for other steps. This is also evident from the Taylor plots of Figure 7 where some “skillful” configurations (e.g. 5 and 10 for GFS rain are not included in your suggested ones. You should also clarify which runs were used for the boxplots (GFS? FNL? Or both?). 

Response 9: The comprehensive performance of each physical scheme can be evaluated by means of a box diagram, regardless of which re-analysis data is used or which physical schemes are combined with this physical scheme. The evaluation index for the box plot is to calculate N1-36 and G1-36, that is, the average metric for all schemes including GFS and FNL. The mean metric of one of the physical scheme is the average of all the combined schemes with this physical parameter. For example, the mean metric of KF CU scheme is the average of the 36 combined schemes including n1-9, n19-27, g1-9 and g19-27. The Taylor graph only represents the performance of the physical scheme in one experiment (WRF-FNL experiment or WRF-GFS experiment). We should combine the performance of each physical scheme in two experiment to judge its simulation effect. Therefore, we finally decided to use this method to evaluate the physical parameter scheme.

Minor comments:

1. Define MP in the introduction.

2. Correct ERI-40 to ERA-40

3. In equation (2) instead of MAE you define RMSE. Please correct.

4. In the first paragraph of the results section you refer to equation (3) for R. I think the correct reference for R is equation (4). In the same section in the text you mention a mean Bias of 0.31 mm, while in Figure 2 this bias is 0.34. Please check again.

Response: Thank you for your advices. We make necessary corrections as follows: 1. Add define MP in the introduction; 2. Correct ERI-40 to ERA-40; 3. Define equation (2) to MAE; 4. Correct reference for R is equation (4), and correct mean Bias of 0.34 mm.

Dear reviewer, we have done a lot of efforts to make our manuscript acceptable. Our revised manuscript is ready for submission. We are waiting desperately for your answer and we hope it will be positive.

Yours sincerely,

Yulin Zhou

The first author

Zhenxia MU

Round  2

Reviewer 1 Report

The revised paper has addressed all my previous comments, and I suggest to accept the paper.

Reviewer 2 Report

The authors have made significant efforts to improve the manuscript. Although the study is very local and  based on very short simulations of a single event, it might be of interest for future WRF studies in similar environments and climate regimes. My recommendation is to accept it in the present form. 

Water EISSN 2073-4441 Published by MDPI AG, Basel, Switzerland RSS E-Mail Table of Contents Alert
Back to Top